# Influence of Foods and Nutrition on the Gut Microbiome and Implications for Intestinal Health

**DOI:** 10.3390/ijms23179588

**Published:** 2022-08-24

**Authors:** Ping Zhang

**Affiliations:** Center for Integrative Conservation, Xishuangbanna Tropical Botanical Garden, Chinese Academy of Sciences, Menglun 666303, China; zhangping@xtbg.org.cn

**Keywords:** gut microbiome, gut microbiota, nutrition, foods, dietary fiber, dietary fats, dietary protein, intestinal health, colitis, IBD

## Abstract

Food components in our diet provide not only necessary nutrients to our body but also substrates for the mutualistic microbial flora in our gastrointestinal tract, termed the gut microbiome. Undigested food components are metabolized to a diverse array of metabolites. Thus, what we eat shapes the structure, composition, and function of the gut microbiome, which interacts with the gut epithelium and mucosal immune system and maintains intestinal homeostasis in a healthy state. Alterations of the gut microbiome are implicated in many diseases, such as inflammatory bowel disease (IBD). There is growing interest in nutritional therapy to target the gut microbiome in IBD. Investigations into dietary effects on the composition changes in the gut microbiome flourished in recent years, but few focused on gut physiology. This review summarizes the current knowledge regarding the impacts of major food components and their metabolites on the gut and health consequences, specifically within the GI tract. Additionally, the influence of the diet on the gut microbiome-host immune system interaction in IBD is also discussed. Understanding the influence of the diet on the interaction of the gut microbiome and the host immune system will be useful in developing nutritional strategies to maintain gut health and restore a healthy microbiome in IBD.

## 1. Introduction

The gut microbiome, or gut microbiota, also termed commensal, refers to the entire microbial community that populates the mammalian gastrointestinal (GI) tract, with the majority residing in the colon. The human gut microbiome reaches 3.8 × 10^13^ microbes in a standard adult male, which outnumbers the human host cells (3.0 × 10^13^) [1]. Each individual hosts at least 160 species out of the total 1150 species that colonize the human GI tract [2]. There are five major phyla for the human gut microbiota, namely *Firmicutes*, *Bacteroidetes*, *Actinobacteria*, *Proteobacteria*, and *Verrucomicrobe*, with the two dominating phyla, Firmicutes and Bacteroidetes, representing 90% of the gut microbiota [3]. Some bacteria termed “pathobionts” can become pathogenic under specific conditions. For example, members of the phylum *Proteobacteria* belong to “pathobionts”, and a bloom of them is seen in inflammatory bowel disease (IBD) [4]. Accumulating evidence has shown that a diversified and well-structured gut microbiota is critical in maintaining health. Dysbiosis, defined as reduced diversity and alterations of the composition of the gut microbiota, is associated with obesity, diabetes, and gastrointestinal diseases such as IBD [2,5,6]. Diet is a driving factor in shaping human gut microbiota composition and function [3,7,8,9,10,11]. There is growing interest in targeting the gut microbiota through diet and nutritional approaches either to promote gut health or as an adjunct therapy for treating IBD [10,12,13].

The human GI tract functions to digest foods and uptake nutrients. It also protects from pathogen infection as well as maintains immune tolerance. Undigested foods reach the colon and serve as substrates for bacterial metabolism. Carbohydrates, proteins, and fats are the three major macronutrients that serve as an energy source in human nutrition; they differ greatly in digestibility and, therefore, provide quite different microbiota-accessible nutrients. The amount and types of macronutrients select the growth of different bacteria and generate different metabolites, which have positive or negative effects on the gut epithelium and mucosal immune system (Figure 1). Indigestible carbohydrates are a major type of dietary fiber and select fiber-degrading bacteria, which produce short-chain fatty acids (SCFAs). SCFAs, in general, are considered to be beneficial to gut health under normal conditions. Undigested proteins in the range of 10–30% promote the growth of proteolytic bacteria, which produce SCFAs, branched-chain fatty acids, and some toxic metabolites, including ammonia and hydrogen sulfides. Bile acids are secreted in response to dietary fats and form conjugated fatty acids. About 5% of conjugated fatty acids reach the colon for bacterial metabolism [14]. Dietary fats select bile acid-tolerant bacteria, which produce toxic compounds like H_2_S [6].

Compared with human metabolism, bacterial metabolism is much more powerful considering the fact that the gut microbial genes (3.3 × 10^6^) far outnumber human protein-coded genes by 150-fold [2]. What is more, bacterial metabolism can switch from one substrate to another substrate much faster, depending on substrate availability. A healthy gut microbiome is characterized by a diversified bacterial community, where different species are equipped with different catabolism capacities and work in concert.

Through generating a diverse array of metabolites, the gut microbiome interacts with the gut epithelium and the intestinal mucosal immune system to maintain gut homeostasis, thus forming a symbiotic relationship with the host. Diet can disturb gut homeostasis by influencing the diversity, composition, and function of the gut microbiome. A nutritionally balanced diet is critical for maintaining a healthy gut microbiome, the integrity of the intestinal barrier, immune tolerance, and normal gut physiology, whereas an unbalanced diet, like the typical western diet, results in reduced diversity and dysbiosis of the gut microbiome, which can lead to a leaky gut and chronic inflammation, as seen in IBD. This review focuses on food and nutrition factors that affect gut health by influencing the interplay of the gut microbiome with the epithelium and intestinal mucosal immune system.

## 2. Diet, the Microbiome, and the Intestinal Barrier

The human GI tract is covered by a single layer of epithelial cells held together by tight junction proteins such as claudins, occludins, and zonulae occudens (ZO) [15]. The intestinal epithelial cells form a physical barrier as they are impermeable to luminal contents. There are at least seven types of intestinal epithelial cells: enterocytes, goblet cells, Paneth cells, microfold cells, enteroendocrine cells, cup cells, and tuft cells [16]. Enterocytes are the most abundant cells responsible for nutrient uptake [17]. Goblet cells, with more abundance in the distal direction, are responsible for producing mucus [17]. Most Paneth cells reside in the small intestine and secret antimicrobial peptides [17]. The intestinal epithelial cells and the secreted factors form the intestinal barrier [15].

The glycoprotein-rich mucus layer overlying the gut epithelium is the first line of defense against commensal microbes as well as pathogens [17]. MUC2 is the major component of the gel-like mucins in the intestine. The large intestine has two layers of mucus, namely, a firmly attached bacteria-free inner layer and a loose outer layer [17]. The inner layer is about 50 μm thick in mice and 200–300 μm thick in humans. The outer layer expands 4–5 times in volume, which creates a habitat for the commensal bacteria. The mucus barrier is also a reservoir of antimicrobial peptides and IgA. The inner mucus layer is continuously renewed every 1–2 h in murine colonic tissue. Once the inner mucus layer is lost or becomes penetrable to bacteria, a large number of bacteria will reach the epithelial cells and trigger inflammation. Thus, a penetrable inner mucus layer allowing large quantities of bacteria to reach the epithelial cells is a common mechanism for all mouse models of colitis and patients with active ulcerative colitis [17].

Bacterial stimulation is essential for the development and function of the intestinal barrier. In germ-free mice, the mucus layer is extremely thin [18]. The permeability of the intestinal barrier is tightly regulated in a healthy gut. The commensal bacteria maintain the epithelial barrier by providing energy in the form of short-chain fatty acids and also releasing antimicrobial substances to inhibit pathogens. Some nutrients are important regulators of tight junction protein levels, which are critical in maintaining the epithelial barrier [15]. An increase in intestinal permeability, termed a “leaky” gut, can be induced by dietary factors and may trigger inflammatory responses [19]. In a healthy gut, a balance exists between commensal bacteria and the mucus layer. Some gut bacteria, termed mucin specialists, specifically metabolize mucins and are the major mucin degraders when the diet is rich in dietary polysaccharides. There is a balance of production and degradation of mucus, which maintains the thickness of the mucus layer. Dietary fiber-derived SCFAs promote the integrity of intestinal epithelium by inducing goblet cells to increase mucin production [2] and enterocytes to secret IL-18, which is important for epithelial repair [20]. SCFAs can also directly modify tight junctions to strengthen the gut barrier [15]. When the diet is devoid of dietary fibers, some mucin generalists switch metabolism from plant polysaccharides to host mucin glycans. Expansion of mucus-degrading bacteria and an increase in the metabolic activity in utilizing mucin glycans lead to erosion of the mucus layer [18]. Reduced dietary fiber correlates with the thinning of colonic mucus. Different protein sources also affect the thickness of the mucus layer [21]. High saturated fats impair intestinal barrier integrity by reducing tight junction protein occludin and ZO-1 [22,23]. Simple sugars [24,25] and emulsifiers [26] negatively affect the intestinal barrier by inducing the expansion of mucin lytic bacteria such as *Akkermansia muciniphila,* which leads to a thinning of the mucus layer. Some food components (milk fat) promote the growth of *Proteobacteria,* which produces compounds that are toxic to the intestinal epithelial cells [26]. A leaky gut is involved in the pathogenesis of many inflammatory diseases, including IBD [19].

## 3. Diet, the Microbiome, and the Intestinal Mucosal Immune System

Underneath the intestinal epithelial layer is the lamina propria, where most of the intestinal mucosal immune system resides [17]. Here, various types of innate and adaptive immune cells are found: dendritic cells, macrophages, innate lymphoid cells (ILCs), CD4^+^ T cells (Th1, Th17, Treg cells), CD8^+^ T cells, and IgA-secreting plasma cells. These cells work in concert in defense against pathogen infection and in the maintenance of the intestinal mucosal barrier. Unrestrained inflammatory responses to food antigens or commensal bacteria are the main causes of chronic intestinal inflammation and tissue damage in human IBD patients [12]. Under normal conditions, the mucosal immune system is tightly regulated. Local Tregs play a critical role in colon homeostasis [27,28]. Many bacterial metabolites induce colonic Tregs, such as SCFAs, certain secondary bile acid conjugates, and tryptophan metabolites [29,30,31,32,33]. The commensal bacteria and the immune system evolve and interplay with each other. Diet influences this interplay by providing substrates for the gut bacteria, and some nutrients can directly modulate immune cells.

Normal development and function of the immune system depend on bacterial stimulation. Germ-free mice show defects in several immune cells and are more susceptible to infection [34]. In mice monocolonized with human gut microbes, immune responses show diversity and redundancy [34]. Most microbes elicit distinct and shared responses at both transcriptional and cellular levels. The broad and redundant immune changes induced by gut microbes provide a consistent impact on the host and promote overall health. A recent human study showed that a diet rich in fermented foods leads to increased microbial diversity and decreases in numerous markers of inflammation [10]. The effect is probably through modulations in the gut microbes and metabolites.

It is well established that Foxp3^+^ Treg cells play a central role in the maintenance of immune homeostasis and particularly in the intestine. This is a subset of CD4^+^CD25^+^ T cells expressing the transcription factor Forkhead box P3 (Foxp3), which could suppress spontaneous multi-organ autoimmunity, including gastrointestinal inflammation induced by CD4^+^CD25^−^ T cells [35]. Tregs represent around 10% of CD4^+^ T cells and were initially discovered to present only in lymphoid tissues; however, recent studies showed the existence of tissue Tregs [36]. Two colonic Treg populations have been identified: one comes from the thymus and proliferates in the colon expressing Helios and Gata3; the other one newly differentiates from naïve Foxp3^−^ CD4^+^ T cells and becomes Helios^−^RORγt^+^ [27]. These colonic Tregs are as effective as lymphatic tissue Tregs in terms of suppression of effector T cells, thus controlling local inflammation. Another distinctive role of colonic Tregs is involved in local mucosal barrier repair. Colonic Tregs have been shown to suppress symptoms in multiple models of colitis [29,30,31,32].

The commensal microbes play a major role in shaping the population of Foxp3^+^CD4^+^ T regulator cells in the colon. Colonic Tregs are reduced in germ-free mice or following antibiotic treatment. A number of individual gut microbes strongly induce colonic Tregs, including *Clostridia* clusters IV, XIVa and XVIII, and some *Bacteroides* species [28]. These bacteria produce SCFAs by fermentation of dietary fiber. The very low number of colonic Tregs in germ-free mice can be rescued by acetate, propionate, or butyrate, indicating these SCFAs work independently [24]. Different SCFAs induce colonic Treg population through multiple mechanisms. For example, acetate promotes the expansion of pre-existing colonic Tregs by activation of FFAR2 on T cells, whereas butyrate increases the de novo differentiation of colonic Tregs by inhibiting histone deacetylase (HDAC) activity [30]. SCFAs also indirectly promote colonic Tregs expansion by affecting DC maturation through activation of GPR109A on DCs [29].

## 4. Foods and Nutrition on the Gut Microbiome and Intestinal Health

### 4.1. Dietary Fiber

#### 4.1.1. Dietary Fiber on Gut Microbial Ecology

The definition of dietary fiber dates back to the 1950s and has evolved in the last several decades. The same basis remains in carbohydrate polymers that are resistant to digestion and absorption in the human small intestine [37]. Although dietary fibers are found in a wide range of plant-based foods such as cereals, legumes, nuts, tubers, vegetables, and fruits, fiber intake is far below the recommended levels in Western countries [37]. Dietary fiber is classified into different types based on chemical structures, including resistant starches (RS), nondigestible oligosaccharides, nondigestible polysaccharides, and chemically synthesized carbohydrates [38]. Nondigestible polysaccharides include cellulose, hemicellulose, polyfructoses, gums and mucilages, and pectins. Not all dietary fibers are fermentable [38]. Cellulose is not fermented by gut microbes and only has bulking effects [39]. Most dietary fibers are fermentable. Several terms are used to define subsets of dietary fiber with respect to their effects on the modulation of the gut microbiome. “Prebiotics” refers to a “nondigestible food ingredient that beneficially affects the host by selectively stimulating growth and/or activity of one or a limited number of bacteria already resident in the colon, and thus helps to improve host health” [38]. As more up-to-date knowledge builds up in terms of the health benefits of dietary fiber on the gut microbiota, the prebiotic concept is becoming considered outdated. A new term “MACs” is introduced as “microbiota-accessible carbohydrates” [40] to stress the capacity of GI microbiota to utilize fermentable dietary fiber.

Dietary fiber is the key nutrient for maintaining the diversity of gut microbiota. Low microbiota diversity is associated with many chronicle inflammatory diseases such as obesity, diabetes, and IBD. A low-fiber, high-fat, high-protein diet is a main contributing factor to the depletion of fiber-degrading microbes in populations in industrialized countries [7]. In healthy humans, a high-fat, high-protein, and low-fat diet leads to reduced gut microbiota diversity as quickly as one day [41]. Lack of fiber can have long-lasting detrimental effects over generations on the gut microbial ecology. A fiber-deficient diet in mice harboring human microbiota results in greatly reduced microbial diversity, and this reduced microbial diversity becomes even worse over generations [42]. The loss of fiber-degrading bacteria species can be rescued in the first generation by adding fiber to the diet; however, over several generations, the depletion is irreversible. In a similar way, the human microbiota shows resilience in response to short-term dietary fiber supplementation (days to weeks) [10]. Dietary intervention with a high-fiber diet for 10 weeks in healthy adults does not increase the gut microbiota diversity but increases carbohydrate-active enzymes (CAZymes).

Besides maintaining the diversity of the gut microbiota, sufficient fiber in the diet helps maintain the integrity of the mucus barrier and eliminates the risk of pathogen infection. A low-fiber diet promotes the expansion of colonic mucus-degrading bacteria, leading to erosion of the intestinal mucus barrier, thus enhancing pathogen susceptibility. Using a gnotobiotic mouse model, Desai et al. [18] showed that, during chronic or intermittent dietary fiber deficiency, mucin-degrading species such as *A. muciniphila* and *B. caccae* increase rapidly with a corresponding decrease in the fiber-degrading species. Consistent with changes in the microbial community abundance, transcriptomic changes also reveal elevated transcripts encoding enzymes to metabolize host sugars in fiber-free diet-fed mice and more abundant transcripts encoding enzymes that target dietary polysaccharides in fiber-rich diet-fed mice. Reduction of mucus thickness by the fiber-free gut microbiota brings luminal bacteria closer to the intestinal epithelium and some host responses, including increased fecal levels of a neutral protein lipocalin, shorter colon length, and changed transcriptomes in immune response pathways in the cecal tissue. When infected with *Citrobacter rodentium*, a murine pathogen that models human enteric *E. coli* infection, fiber-deprived microbiota promotes greater access to the pathogen, wider areas of inflammation in colon tissue, and lethal colitis. Importantly, adding purified soluble fiber to the fiber-free diet does not alleviate the degradation of the mucus layer, highlighting the importance of complex plant fiber with intact plant cell walls in maintaining a functional mucus layer.

#### 4.1.2. Dietary Fiber Fermentation by the Gut Microbes Is Type Specific

Since dietary fiber is a chemically heterogeneous group of molecules with different structures and different physical forms, many factors affect microbial utilization of dietary fiber, including source, chain length, sugar types, linkages types, particle size, and association with other compounds [39]. Although there is only a small number of the composing monosaccharides (glucose, galactose, mannose, fructose, arabinose, xylose, rhamnose, fucose), a wide range of structures exist due to combinations of different linkages between two sugar moieties as well as the next-level branching units [39]. Since the bacteria is specific at cleaving different linkages of carbohydrate polymer, chemically different dietary fibers are selectively used by different gut bacteria equipped with different degrading enzymes. Cellulose with a structure of β-(1,4)-linked poly-D-glucose is generally not fermentable, although cellulose-degrading bacteria are identified in the human gut [39]. Resistant starch (α-(1,4)-linked glucose with α-(1,6)-linked glucose branches), pectin (α-(1,4)-linked D-galacturonate), and inulin (β-(2,1)-linked fructose) are all fermentable and associated with distinct gut microbiome compositions [39]. Even subtle variations in chemical structures of arabinoxylan (AX) isolated from three wheat classes influence the composition and function of gut microbiota [43]. The AX with a shorter backbone and more branches leads to much higher diversity and evenness compared with two AXs with a longer backbone and fewer branches. The AX with a shorter backbone is favored by *Bacteroides*, whereas the AXs with longer backbones and fewer branches are favored by *Prevotella*. Both *Bacteroides* and *Prevotella* are degraders of arabinoxylans. *Prevotella* competes better for fewer-branched AXs, but *Bacteroides* compete for more branched AXs.

The gut microbes rely on not only carbohydrate-active enzymes in cleaving the sugar linkages but also carbohydrate-binding proteins and transporters to colonize around fiber particles. Thus, physical forms of dietary fiber, including factors such as fiber matrix and particle size, determine the easiness of rapid colonization of gut bacteria and subsequent cleavage of the linking chemical bonds. Gut bacteria must first attach to dietary fiber particles to use the substrate. Different bacteria show varied abilities in attaching the same dietary fiber. Thus, dietary fiber, with its distinct physical structure, determines bacteria specificity to gain access and further utilization.

Three type-IV resistant starches (maize, potato, or tapioca derived) with small differences in chemical structure and granule size are shown to distinctively affect the gut microbiome in healthy humans compared with native corn starch, a high-amylopectin starch that is rapidly digested and absorbed in the small intestine [9]. The maize RS4 is produced through an annealing and acid treatment of high-amylose maize starch (with restructured starch granules), whereas the potato RS4 and tapioca RS4 are produced by phosphate cross-linking the native starches (with inter-starch ester linkages). Four weeks of consumption of maize and tapioca RS4s increases interpersonal variation, shifts community composition, and reduces community evenness. The potato RS4 does not affect gut microbiome diversity. In terms of fecal microbiota composition, potato RS4 behaves like corn starch. In contrast, both maize RS4 and tapioca RS4 alter the relative abundance of distinctive taxa. The maize RS4 selectively enriches *B. adolescentis*, *E. rectale*, *Oscillibacter* spp., and *Ruminococcus* spp., while tapioca RS4 selectively enriches the family *Porphyromonadaceae*, the genus *Parabacteroides*, and *Parabateroids distasonis*, *Parabateroids* spp., *Faecalibacter prausnitzii*, and *Eisenbergiellia* spp. In addition, both maize and tapioca RS4 enrich *Bifidobacterium adolescentis*. Moreover, maize RS4 reduces *Ruminococcus callidus*, *Agathobaculum butyricproducens*, and *Adlercreutzia*, and tapioca reduces an unclassified genus of *Ruminococaceae* related to *Eubacterium hallii* and *Clostridium viride*.

The substrate specificity of chemically modified RS4 is attributed to selective bacterial adherence. An in vitro experiment has shown that *E. rectale* only adheres to maize RS4 but not potato or tapioca RS4, while *B. adolescentis*, *R. bromii*, *P. distasonis* are able to adhere to all tested RS4s. *E. rectale* is completely unable to bind tapioca RS4. *B. adolescentis* and *P. distasonis* show good adherence and utilization of tapioca RS4. Phosphate crosslinking of starch granules generates inter-starch ester linkages that impede attachment of *E. rectale* and cross-feeding bacteria, conferring competitive advantages to *P. distasonis*. *B. adolescentis* is the only species that is able to use both crystalline and cross-linked starches.

Processing can influence the particle size of dietary fiber and thus bacterial utilization. Different-sized cereal bran fractions influence the composition and metabolic function of gut microbiota. The chemical composition of maize bran can be influenced by particle size [44]. The smallest size fraction (180–250 μm) has higher glucose, mannose, and galactose and lower arabinose and xylose contents compared with those of larger sizes (250–300 μm, 300–500 μm, 500–850 μm). Consequently, the smallest size fraction produces much higher SCFAs from 6–48 h in an in vitro fermentation system inoculated with human fecal microbiota. While small particles favor the families *Ruminococcaceae* and *Porphyromonadaceae*, medium particles favor the family *Bacteroidaceae*. Similar to maize bran, fine-ground wheat bran (FWB, 438 μm particle size) has higher soluble fiber, swelling capacity, water-holding capacity, and fermentability than coarse wheat bran (CWB, 605 μm particle size) [45]. When feeding pregnant sows, FWB resulted in a significantly increased abundance of *Bacteroidetes* and decreased abundance of *Firmicutes* at the phylum level. FWB also markedly increased total fecal SCFAs.

Plant source of dietary fiber is an important determining factor for gut bacterial fermentation due to varied physiochemical properties. For example, maize bran has much lower fermentability compared with wheat bran. Supplementation with maize RS4 in healthy humans for four weeks selectively increases butyrate concentrations, while tapioca RS4 increases propionate compared with digestible corn starch [9]. Potato RS4 does not change individual SCFAs, indicating potato RS4 is not fermented in the colon [9].

Different types of dietary fiber vary greatly in terms of fermentability and therefore produce quite different metabolic and immunological consequences in the host. For example, carboxymethylcellulose (CMC), a synthetic fiber commonly used in research rodent diets, is not fermentable. Mice fed a high-fat diet containing CWC gained weight rapidly and were associated with a high *Firmicutes*/*Bacteroidetes* (F/B) ratio [46]. In contrast, the same amount of bamboo shoot fiber, an insoluble fiber rich in fermentable hemicellulose, almost completely suppressed high-fat diet-induced weight gain and counteracted all the effects of high fat on the gut microbiota, including reduced diversity and F/B ratio and detrimental effects on metabolism [46]. Therefore, bamboo shoot fiber can be considered an MAC, conferring beneficial effects in the context of a high-fat diet. Surprisingly, the same amount of inulin had little effect on high-fat diet-induced body weight gain. Similarly, wheat bran and soy fiber also promoted weight gain as CMC in the high-fat diet context. Not all soluble fibers are similar in their physiological effects [47]. For example, both inulin and pectin are rapidly fermented, but they have contrasting effects on colitis. In αIL-10R-induced colitis in mice, compared with the cellulose control, pectin attenuated colonic inflammation, decreased abundance of *Proteabacteria*, enhanced gut barrier function, and increased colonic Tregs. On the contrary, the same amount of inulin exacerbated colonic inflammation, increased levels of *Proteobacteria* and SCFA-producing bacteria, including *Clostridia* cluster XIVa, *Lachnospiraceae*, and *Ruminococcaeae*, and worsened gut barrier dysfunction [48]. Suppressing butyrate production mitigated colonic inflammation in inulin-fed mice, suggesting a role of excessive butyrate produced by inulin-exacerbated colitis. More research is needed to understand the mechanisms of the differential modulation of the gut microbiota by different dietary fibers on host metabolism and immune function.

### 4.2. Dietary Fat on Gut Microbiome 

A high-fat diet leads to reduced diversity of the gut microbiota in humans and rodents compared with low-fat diets, regardless of the fat source being lard, milk, safflower oil, or palm oil [46,49,50,51]. Different dietary lipids have differential effects on the composition of gut microbiota due to their difference in fatty acid profiles. In rodents, a high amount of lard-based saturated fat increased the ratio of *Firmicutes*/*Bacteroidetes* [49,50], induced a reversible bloom of *Firmicutes* class *Mollicutes* [49] or *Firmicutes* class *Clostridiales*, and a decrease in *Bacteroidetes* class *Bacteriodles* [50]. Palm oil strongly increased *Verrucomicrobia* (due to increased abundance of *Akkermansia muciniphila*) at the phylum level and increased the *Firmicutes*/*Bacteroidetes* ratio [46]. All these microbial changes promote obesity. A supplement of 0.5% w/w conjugated linoleic acids (CLAs) in the regular mice diet was shown to influence the ratio of *Firmicutes*/*Bacteroidetes* at the phylum level [52]. Omega-3 PUFA supplementation in healthy middle-aged volunteers with a daily 4 g dose induced a reversible increase in the abundance of several genera, including *Bifidobacterium*, *Roseburia* and *Lactobacillus* but had no effect on microbial diversity and no taxa changes at the phylum level [53]. High dietary milk fat (rich in saturated fatty acid), but not safflower oil (rich in n-6 polyunsaturated fatty acids), induced a bloom of a sulfate-reducing microbe *Bilophia wadsworthia* [51]. Milk fat increased the incidence of colitis with typical Th1 inflammatory response in genetically susceptible IL10^−/−^ mice but not in wild-type mice. These effects are mediated by a milk-fat-induced increase in taurine-conjugated bile acids [51]. Thus, by influencing the bile acid metabolism, a high intake of certain dietary fat promotes pathobiont expansion, which leads to intestinal inflammation in a genetically susceptible host. Altered bile acid pool composition is associated with IBD [2]. Besides diet, bile acid pool size also depends on the gut microbes. About 5% of bile acids in the colon are transformed into secondary bile acids, and some secondary bile acids are known to induce colonic Tregs, which control gut inflammation [31,32]. Although n-3 fatty acids show some protective effects in animal models of colitis, their benefits in human IBD remain inconclusive [54]. Some dietary fibers are very effective in counteracting high-fat diet-induced alterations in the gut microbiota and related metabolic or immune dysfunction [46,48]. Future research is needed to address the question of how n-3 fatty acids, CLA, and dietary fibers affect bile acid metabolism. Further exploring the complex regulation of the bile acid pool by different nutrients and bacterial metabolism of bile acids will help better understand the pathology of IBD and establish proper nutritional advice on dietary fat and dietary fiber for IBD patients.

Fat-soluble vitamin D and vitamin D receptors play critical roles in the regulation of gut microbiota and immune responses and have a protective role in IBD [55]. Dietary vitamin D or that synthesized in the skin is converted to the active form 1, 25-dihydroxyvitamin D_3_ (1, 25(OH)_2_VD_3_), which is the primary ligand for vitamin D receptor (VDR), which is highly expressed in the small intestine and colon. Vitamin D promotes barrier function by upregulation of the expression of tight junction proteins ZO-1, ZO-2, and claudin-2 [56]. Vitamin D_3_ metabolite 1, 25(OH)_2_VD_3_ induces Foxp3 expression Tregs from human peripheral CD4^+^ T cells. VD_3_-induced Tregs can suppress the proliferation of CD4^+^ T cells in a cell-contact-dependent manner [57].

Low vitamin D status has been observed in IBD patients [55]. Diet can only provide 10–20% vitamin D; about 90% of vitamin D is synthesized in the skin through ultraviolet B light [56]. Not having enough sun exposure is a major cause of vitamin D deficiency. Vitamin D deficiency in mice led to reduced microbial diversity, altered composition of the gut microbiota, and susceptibility to colitis [58]. VDR dysfunction also plays a role in IBD. *Lactobacillus* is depleted, and *Clostridium* and *Bacteroides* are increased in VDR KO mice [59]. *Parabacteroides* is found to be the most significant taxon correlated with the human Vdr gene [60]. The VDR KO mice had impaired colonic antibacterial activity and were predisposed to colitis. IL-10 knockout mice have lower intestinal VDR expression and develop spontaneous colitis. VDR-IL10 double knockout mice develop even more severe colitis than IL-10 knockout mice [55].

### 4.3. Dietary Protein and Certain Amino Acids

Protein intake in a typical Western diet is about 1.2–1.4 g/kg, which is well in excess of the recommended level of 0.6–0.8 g/kg/d [61]. Some high-protein diets aimed at weight loss recommend even higher protein intake, typically 25–35% of energy intake [62]. Dietary protein is not 100% digested. Digestibility of animal protein exceeds 90%, and plant protein digestibility is in the range of 70–90%. On a normal mixed diet, the amount of protein rather than its source determines the amount reaching the colon [63]. Undigested protein is mainly fermented in the distal colon and produces far more complex metabolites than carbohydrate fermentation, including SCFAs, branched-chain fatty acids, carbon dioxide, hydrogen, hydrogen sulfides, ammonia, phenols, and indole derivatives [64]. Some of these metabolites (hydrogen sulfides and ammonia) are toxic compounds and could be potentially detrimental to colonic epithelium at excessive amounts [63], while others (indole derivatives) are essential for the expression of IL-22, a cytokine that supports the integrity of intestinal mucosa [2].

The quantity and source of dietary protein determine the amounts and profile of bacterial metabolites [65]. A high-protein diet shifts gut bacteria metabolism to protein fermentation and can disturb the gut mucosal homeostasis. A 3-week human dietary intervention study in overweight humans showed that high-protein diets with casein or soy protein as the protein source do not alter the gut microbiota composition but induce a gut bacterial metabolism shift towards amino acid catabolism with different metabolite profiles [65]. Casein and soy protein show specificity in regulating gene expression involved in rectal mucosal homeostasis, such as the cell cycle and cell death [65]. High animal protein intake is associated with an increased risk of IBD [6,11,63]. Processed meat contains high amounts of sulfated amino acids, which are fermented by sulfate-reducing bacteria to generate hydrogen sulfide (H_2_S); therefore, processed meat is excluded in the Crohn’s Disease Exclusion Diet, which has proven to be successful in achieving clinical remission [63]. Similar to human studies, dietary methionine restriction in high-fat diet-fed mice showed an altered gut microbiota, improved intestinal permeability, and reduced inflammation [66]. A 6-week high-protein diet (HPD, 45% protein) in adult Wistar rats leads to a significantly altered gut microbiome composition characterized by the increased relative abundance of *Bacteroidetes*, decreased relative abundance of *Firmicutes*, *Actinobacteria*, and *Acidobacteria* at the phylum level, highly increased *Escherichi*/*Shigella*, *Enterococcus* at the genus level and decreased *Ruminooccus bromii*, *Akkermansia muciniphila* at the species level compared with a normal protein diet [67]. Consistent with decreased propionate- and butyrate-producing bacteria, concentrations of acetate, propionate, and butyrate are decreased by HPD [68]. A high-protein diet leads to increased levels of unhealthy microbial metabolites represented by spermine, cadaverine, and sulfide [67]. The colon epithelial transcriptomes are also extensively altered by an HPD with upregulation of many genes involved in chemotaxis, the TNF-α signal process, and apoptosis and the downregulation of genes involved with immunoprotection. Long-term (24-week) feeding of mice with a high-protein (52% energy from casein compared with 20% energy from casein in the control) diet resulted in decreased intestinal occludin gene expression, increased plasma endotoxin and monocyte chemoattractant protein-1, indicating a leaky gut and systemic inflammation [61].

Although numerous studies reported various protein sources differentially affect the composition of the gut microbiota [69], only a few investigated their effects on gut mucosal barrier function. A study in mice showed that both chicken and soy protein in a formulated purified diet induced a shift from *Bacteroidetes* dominance (regular chow) to *Firmicutes* dominance of the gut microbiota at phylum level following 4 weeks of feeding [2]. Chicken meat also results in a higher abundance of the phyla *Actiobaceria* and *Verrucomicrobeia*, whereas soy protein increases the abundance of *Proteobacteria*. At the genus level, chicken meat better supports the growth of *Akkermansia muciniphila* than soy protein. Chicken meat induces a higher number of goblet cells and a thicker mucus layer, suggesting that at recommended levels, chicken meat is better than soy protein at maintaining the gut mucus barrier. The gut microbiota in rats also shows a distinct response to beef, chicken, and soy protein in the diet [2]. Long-term (90-day) soy protein intake in growing rats leads to increased mRNA levels of LBP and CD14 in the liver compared with casein, beef, and chicken as protein sources, indicating increased levels of bacterial endotoxins [70]. However, plant proteins may confer some beneficial effects in the context of a Western diet. Three soy protein preparations increased the gut microbial diversity in hamsters fed with a Western diet for 6 weeks compared with the same amount of milk protein and altered the gut microbial composition at all taxonomic levels, which was associated with reduced lipogenesis [71]. The direct effect on gut barrier function was not measured in this study. Differential effects of soy protein and casein on bile acid metabolism and different antibacterial peptides composition between soy protein and casein may explain the diet-induced changes as they are likely to affect colonic Tregs and intestinal homeostasis.

A novel protein source is promising in promoting a healthy microbiome and gut health. A recent study showed that replacing the protein source in a high-fat, high-sugar Western diet from casein to whole-cell lysates of the non-commensal bacterium *Methylococcus capsulatus* Bath (McB) reverses the gut microbiota to a structure more resembling that in low-fat diet-fed mice [72]. WcB-induced gut microbiota changes include a significantly lower F/B ratio, a reduction in the obesity-related genus *Desulfovibrio*, and a bloom of the *Parasutterella* and *Parabacteroides* genera. An McB diet also induces more SCFAs in the cecum and proximal colon. Besides its beneficial effects on metabolic parameters, McB also induces FoxP3^+^RORγt^+^ colonic Tregs and markedly enhances neutral mucins production by goblet cells and mucin glycosylation status, indicating its direct role in improving intestinal health.

In humans, protein malnutrition has long been known to be associated with immune defects and intestinal inflammation [73]. In mice, a protein-free diet worsens DSS-induced colitis [74]. The essential amino acid tryptophan is a key regulator of gut immunity. A tryptophan-free diet induces much more weight loss in TNBS-induced colitis mice [74]. Dietary supplement of tryptophan in mice confers resistance to DSS-induced colitis [75]. Tryptophan and its microbial metabolite nicotinamide directly regulate intestinal epithelial immunity and gut microbiota [74]. The angiotensin-converting enzyme (ACE) 2 associates with the apical amino acid transporter B^0^AT1 to regulate the uptake of tryptophan. Mice lacking *Ace*2 have markedly decreased intestinal uptake of tryptophan and develop more severe colitis than wild-type mice. Dietary supplementation of tryptophan or nicotinamide induces antimicrobial peptides in the small intestine and rescues *Ace*2 mutant mice from severe colitis. Interestingly, both tryptophan and nicotinamide reverse the altered luminal ileocaecal microbiome of *Ace*2 mutant mice to be more similar to that of wild-type mice.

Dietary tryptophan is metabolized by host and gut microbes into several indole derivatives, which also act as the aryl hydrocarbon receptor (AhR) ligands [2]. The AhR plays a central role in intestinal mucosal homeostasis. AhR is required by IL-22 production by the type 3 innate lymphoid cells (ILC3s). Il-22 is important for intestinal integrity by inducing antimicrobial peptides from epithelial cells and mucin production by goblet cells, thus conferring protection from pathogen infection and inflammation. *Ahr*^−/−^ mice develop more severe DSS-induced colitis than wild-type mice. Administration of indole-3aldehyde (IAld) rescues wild-type mice from DSS-induced colitis but not the *Ahr*^−/−^ mice, indicating the role of AhR in mucosal protection [76]. Mice deficient for Card9, a susceptibility gene for human IBD, are associated with decreased IL-22 and are more susceptible to colitis [33]. Gut dysbiosis in Card9-deficient mice is associated with decreased AhR activation and decreased bacteria-derived tryptophan metabolites, which resembles human IBD patients. The endogenous metabolite kynurenine is generated by IDO activity in DCs [77]. Kynurenine, at concentrations comparable to levels seen in areas of inflammation, binds to the AhR on naïve CD4^+^ T cells and leads the cells to differentiate into FoxP3+ Tregs, which function to suppress the immune response [78].

Another tryptophan derivative, niacin (vitamin B3), indirectly promotes colonic Treg differentiation by binding GPR109A (encoded by *Niacr*1) on colonic DCs and macrophages [78]. *Niacr*^−/−^ mice are more susceptible to colonic inflammation induced by azoxymethane (AOM) and DSS. Antibiotic treatment results in the aerobic and anaerobic bacteria counts decreasing to less than 1/300 in wild-type mice with reduced butyrate production and activation of GPR109A. Antibiotic treatment aggravates DSS-induced colitis. Administration of niacin in drinking water ameliorates colitis symptoms, suggesting the role of GPR109A in colonic health. Under conditions of decreased butyrate production in the colon, pharmacological doses of niacin might be effective in protecting against colon inflammation by maintaining GRP109 signaling.

L-serine catabolism has a minimum role in a healthy gut. However, during intestinal inflammation, the ability to utilize L-serine confers *Enterobacteriaceae* growth advantage against their competitors and thus plays a key role in the pathogenesis of *E. coli*-driven colitis. In CD microbiota-driven colitis, dietary restriction of L-serine reduces gut colonization of pathogenic *Enterobacteriaceae*, thus attenuating colitis [79].

### 4.4. Food Components Negatively Influence the Gut Microbiome and Intestinal Health

Some minor food components, such as polyphenols [80] and the micronutrient selenium, promote the growth of beneficial bacteria species [81], whereas many others, such as certain food additives and alcohol, can influence the intestinal barrier negatively and lead to a leaky gut which potentially facilitates the translocation of a large number of gut bacteria, pathogen infection, and subsequent gut inflammation [82].

High sugar intake is a hallmark of the Western diet. The link between high sugar intake and IBD was recently demonstrated in animal models of colitis. Short-term intake of high glucose or fructose does not trigger inflammatory responses in the gut of healthy mice but aggregates colitis in DSS-treated mice or IL-10^−/−^ mice; both are known to have a leaky gut [24]. High dietary sugars markedly induce an increase in the abundance of the mucus-degrading bacteria *Akkermansia muciniphila* and *Bacteroides fragilis*, which leads to erosion of the colonic mucus layer. Sugar-induced exacerbation of colitis was not observed when mice were treated with antibiotics or maintained in a germ-free environment, suggesting that the effect was mediated by alteration of gut microbiota. Long-term feeding of C57BL/6 mice with a high-glucose or high-fructose diet leads to reduced microbial diversity and changes in the composition of gut microbiota characterized by a decreased relative abundance of *Bacteroidetes* and increased relative abundance of *Proteobacteia* at the phylum level compared with normal chow diet [25]. Moreover, high glucose or fructose triggers gut inflammation, which leads to increased gut permeability due to decreased expression levels of tight junction proteins. These studies suggest that a high-sugar diet might promote gut microbiota dysfunction, gut barrier integrity, and IBD development.

Food additives are widely used in processed foods produced by the modern food industry. Although they are generally regarded as safe, recent studies suggest they may contribute to the increased incidence of metabolic syndrome and inflammatory diseases by negatively influencing the gut microbiota [82]. For example, two commonly used emulsifiers, carboxymethylcellulose (CMC) and polysorbate-80 (P80), administered in mice at relatively low concentrations, resulted in decreased mucus thickness by more than half and microbiota encroachment [26]. CMC and P80 did not change the total levels of faecal bacteria but dramatically changed the microbiota composition, including reducing *Bacteroidles* and increasing *Ruminoccus gnavus,* which is mucolytic. In IL-10^−/−^ mice, both CMC and P80 induced a marked reduction of microbial diversity, highly increased *Verrucomicrobial* phyla (*Akkermansia muciniphila*), and increased *Proteobacteria*. Emulsifier-treated mice have much higher colitis incidence and severity, indicating the emulsifiers increase the pro-inflammatory potential in susceptible subjects.

## 5. Dysbiosis in IBD and Gut Microbiome-Targeted Therapies

Inflammatory bowel disease (IBD) is a chronic and relapsing disorder of the gastrointestinal tract which is characterized by debilitating relapsing and remitting intestinal mucosal inflammation [83]. Two major types are ulcerative colitis (UC) and Crohn’s disease (CD). The precise etiology of IBD is not completely understood; multiple factors, including genetics, host immune dysregulation and environmental factors, are involved. Current theories believe that alterations of the gut microbiome trigger aberrant immune responses in genetically susceptible individuals, leading to chronic mucosal inflammation [12]. The prevalence of IBD in Western countries is high and rising in developing countries with ongoing industrialization [13]. Diet plays a pivotal role in the pathogenesis of IBD. The Western diet is characterized by high intakes of red meat, butter, dairy products, refined grains, sugar drinks and a lower intake of fruits and vegetables. In contrast, the generally considered healthy Mediterranean diet, characterized by a high intake of fruits and vegetables, whole-grain cereals, poultry and fish, is proven to promote diversity and richness of the gut microbiota [11]. A Western diet is identified as a risk for IBD pathogenesis by epidemiological studies [12,13]. Specific dietary components such as saturated fatty acids, animal protein, and simple sugars may promote dysbiosis of gut microbiota, increase the mucosal barrier, and promote inflammation. Other dietary components, such as docosahexaenoic acid (DHA) and high total fiber intake, are associated with a lower risk of IBD [13].

Gut dysbiosis is observed in a subset of human IBD patients [4,83]. The IBD microbiota has reduced alpha diversity and compositional changes. A decrease in the relative abundance of *Firmicutes* and *Bacteroidetes* and an increase in the relative abundance o*f Enterobacteriaceae,* including *Escherichia coli* and *Fusobacterium,* are the common features [4,83]. Chronic inflammation in the IBD gut leads to meta-transcriptome changes in the gut microbiota adapted to oxidative stress indicating functional changes in the microbial community [84,85]. In parallel with reduced microbial diversity, less diverse and large-scale dysregulated metabolite pools are observed in IBD patients [83]. The IBD gut metabolomes feature reduced SCFAs, vitamins B5 and B3, secondary bile acids, increased amino acids, branched amino acids, branched-chain fatty acids, acylcarnitine, cholate, chenodeoxycholate, and taurochenodeoxycholate [83,85]. The role of gut microbiota in IBD etiology is demonstrated by its pro-inflammatory properties. Transferring IBD microbiota induces more severe colitis in IL-10^−/−^ mice [79] and enhances naïve CD4^+^ T cell-induced colitis severity in *Rag*1^−/−^ mice [86]. The gut microbiotas from healthy human donors and IBD patients have differential immunomodulatory impacts on the homeostatic intestinal T cell response despite similarities in composition and diversity. Germ-free mice colonized with IBD microbiotas have more gut Th17 and Th2 cells and decreased numbers of gut RORγt^+^ Tregs compared to mice colonized with microbiotas from healthy donors [86].

Restoring the gut microbiome composition and function in IBD through diet, prebiotics, antibiotics and fecal microbiota transplantation (FMT) has been investigated [12]. Several diets have proven to have positive effects, including exclusive enteral nutrition (EEN), the specific carbohydrate diet (SCD), the Crohn’s Disease Exclusion Diet (CDED), and the low FODAMP (fermentable oligosaccharides, monosacchardies, disaccharides and polyols) diet [12,13]. A low-fat, high-fiber diet led to reduced inflammation and improved quality of life in patients with ulcerative colitis compared with an improved standard American diet [87]. The difference in macronutrients induced microbial changes at all taxonomy levels, with a significant increase in the relative abundances of *Bacteroidetes* at the phylum level and *Prevotella* at the genus level. Increased dietary fiber intake is partly responsible for the benefits of UC in remission. However, fructans, but not galacto-oligosaccharides or sorbitol, are shown to exacerbate gut symptoms, including the severity of pain, bloating, and flatulence in quiescent IBD patients, indicating differential impacts of FODMAPs on gut physiology depending on type and quantity [88]. Restriction of FODMAP intake can relieve symptoms of IBD without affecting gut microbiome diversity and inflammation markers despite a decline in *Bifidobacteria* and *F. prausnitzii* abundance and lower concentrations of SCFAs [89]. The detrimental effects of high amounts of FODMAPs on gut health have been demonstrated in many animal studies, and high concentrations of SCFAs produced from rapid fermentation are the main mechanism for epithelium injury in the inflamed colon [90]. In contrast to FODMAPs, many plant polysaccharides with distinctive complex structures from various edible plant foods, including tubers [91,92,93,94], bamboo shoots [95], fruits [48,96], and mushrooms [97,98,99], have been proven to promote gut health. In animal models of colitis, these plant polysaccharides are able to reduce mucosal inflammation, strengthen the intestinal barrier, increase SCFA production, correct dysbiosis, and modulate T cells (Table 1). In most of these studies, the soluble fibers are used in a preventative way. Only the mannoglucan from Chinese yam can heal diseased colitis mice [92]. Future studies are required to verify these plant polysaccharides in the treatment way in animal models of colitis and their usefulness in IBD patients. In view of the more complicated diet-gut microbe interaction in IBD, combined therapy of Treg-producing bacteria *Clostridia* and *Bacteroides* with certain polysaccharides supplementation with current drug therapy will likely yield better results in treating IBD.

## 6. Future Perspectives and Conclusions

The human gut microbiome varies in each individual due to differences in type of delivery, infant feeding methods, age, as well as medications such as antibiotic treatment [100]. Considering the huge influence of food components and nutrients on the gut microbiome, future investigations need to examine their roles in more dysbiosis-related intestinal disorders beyond IBD, for example, antibiotic-related adverse effects and gastrointestinal cancers. Moreover, their roles in the pathology of IBD may be investigated from a broader perspective and by a more integrative approach, such as the new paradigm widely applied in cancer research, namely molecular pathological epidemiology (MPE), which incorporates molecular pathology into epidemiological research [101].

Antibiotic treatment is a common medical practice for life-threatening bacterial infections, various surgeries, as well as microbe-associated cancer [102]. However, indiscriminate use of antibiotics has some negative impacts on the gut microbiome. Firstly, it leads to the emergence of resistant strains and enrichment of antimicrobial-resistant genes, which could be rapidly transferred to the surrounding microbes through horizontal gene transfer [103]. Secondly, it disrupts the commensal microbiome structure and reduces some essential bacterial metabolites, leading to impairment of the host proteome [104].

Recovery of the human microbiome following antibiotic treatment is very slow and incomplete. It may take years for full recovery [105]. Probiotics are currently widely prescribed for the prevention of negative impacts on the gut microbiome and related adverse effects post-antibiotic treatment. However, probiotics are shown to impair mucosal microbiome reconstitution and host transcriptome recovery in healthy humans following 7 days of antibiotic treatment [106]. In contrast, autologous fecal microbiome transplantation (FMT) achieves rapid and nearly complete recovery of the mucosal microbiome and gut transcriptome [106].

The great success of FMT, where the microbiome from healthy donors is transplanted to a recipient, in treating recurrent *Clostridium difficile* infections [107] has spurred interest in its use in various clinical contexts, including metabolic disease, IBD, and cancer [108,109,110,111]. However, FMT confers a risk of transferring antibiotic-resistant strains; therefore, quality control is critical for therapeutic microbiota-based drugs [112]. Clinical trials of FMT in treating IBD have produced mixed results [12]. Additional studies are required to establish the conclusive efficacy of FMT in treating IBD. A purified bacterial cocktail consisting of better safety profiles from healthy donors has achieved success in treating *C. difficile* infection [113] and enhancing anti-tumor drug efficacy [114]. Dietary approaches, such as certain types of dietary fiber, can be investigated for post-antibiotic microbiome recovery. For example, bamboo shoot fiber has been shown to strongly inhibit *Verrucomicrobia* [46], which is known to bloom following antibiotic treatment.

Accumulating evidence supports the role of dysbiosis of the gut microbiome in the development and progression of colorectal cancer (CRC) [115], which is the second leading cause of cancer-related death worldwide. Particularly, an increased amount of *Fusobacterium nucleatum* was identified in tumor tissues and fecal specimens from CRC patients [115]. Evidence also suggests that microbiota is genetically regulated in CRC. Genome-wide genotyping of CRC patients identified a number of germline variants associated with the abundance of the genera *Bacteroides, Ruminococcus, Akkermansia, Faecalibacterium* and *Gemmiger* and with alpha diversity [116]. These variants may regulate the microenvironment to favor bacteria growth, therefore modifying disease phenotype by gene-by-environment interactions. Whole exome sequencing analyses also find gene-microbiota interactions in IBD [117] which suggests that genetic variants associated with microbiota also affect the immune system.

Cancer is currently recognized as a microenvironmental, systemic and environmental disease, presenting new opportunities for transdisciplinary microbiomic studies [115]. This is also true for IBD. As described in previous sections, diets and specific food components influence the microbiome, gut epithelium and mucosal immune system. Diets also interact with other lifestyle factors such as exercise and sleep patterns. These factors may influence molecular pathologies in each patient differentially. Diets can be analyzed not only in relation to the incidence (or course) of disease but also in relation to pathogenic mechanisms in diseased tissue. In most current IBD research, dietary factors and the gut microbiome are separately studied in relation to disease or foods/nutrients in relation to stool microbes. These piecemeals can be integrated to obtain better insights into the disease etiology, as has been done in MPE studies in cancer, which integrated analyses of the exposure, microbiome and tumor microenvironment [115,118]. This will be a promising direction for researching dietary factors, microbiome, and personalized biomarkers in IBD. The widely open opportunity is to examine microbes and immune cells in the gut epithelium microenvironment as affected by foods and nutrition. In this era of big data health science, large population-based genomics, metagenomics, and multi-omics (epigenomics, transcriptomics, proteomics, and metabolomics) on the host and gut microbiome can be available, and MPE studies make it possible to integrate data of host genetics and modifiable factors such as diet into the analysis of the microbiome and tissue characteristics, thereby contributing to precision medicine and prevention.

Clearly, a nutritionally balanced, whole food-based fiber-rich diet not only provides the host with all necessary nutrients but also nourishes a healthy gut microbiome with high diversity and well-balanced composition. Many ingredients in modern food processing have detrimental effects on the intestinal barrier and lead to reduced diversity and compositional changes in the gut microbiota, which promote obesity and may predispose to intestinal inflammation in susceptible subjects. Food choices must take the gut microbiome into account for intestinal and overall health, and this is particularly important for IBD or CRC patients. As shown by a diet intervention in IBD patients, a low-fat, high-fiber diet increased fecal tryptophan levels compared with a high-fat diet [87], and macronutrients can have a strong impact on the gut microbiome and metabolites from other macronutrients. Certain dietary fibers can effectively counter the effects of a high-fat diet on the gut microbiome and should be explored as functional ingredients for healthy meat product development by the food industry [46,119]. More research is needed to look at diet-gut microbe interactions in health and the molecular mechanisms of how dietary components or nutrition status contribute to intestinal immune homeostasis. As the effects of more and more novel dietary components and their microbial metabolites on the gut microbiome are discovered by basic research, their usefulness in improving gut health and in the prevention and treatment of gastrointestinal diseases remains to be clarified by human population studies using the MPE approach.

## Figures and Tables

**Figure 1 ijms-23-09588-f001:**
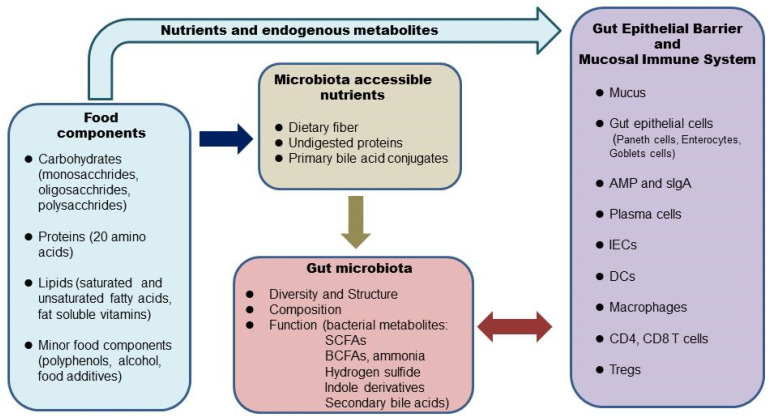
Impacts of foods and nutrition on the microbiota-host interactions in the gut. The arrow indicates regulation. Food components and endogenous metabolites of nutrients directly modulate the gut epithelial barrier and mucosal immune system. Diet also determines microbiota-accessible nutrients, which play a critical role in the gut microbiota ecology. The interaction between the gut microbiota with host epithelium and the mucosal immune system determines intestinal homeostasis. IEC, intraepithelial lymphocytes; AMP, antimicrobial peptides; sIgA, secretory immunoglobulin A; DCs, dendritic cells; SCFAs, short-chain fatty acids; BCFAs, branched-chain fatty acids.

**Table 1 ijms-23-09588-t001:** Beneficial effects of plant polysaccharides in animal models of colitis.

Polysaccharide	Food Source	Animal Model and Experimental Design	Changes in the Gut Microbiome and Inflammation	Reference
CYP-1	*Chinese yam*, the rhizome of *Dioscorea opposite* Thunb	DSS-induced colitis in C57BL/6 mice 3% DSS treat 1 wk followed by CYP-1 for 7 d	↑mRNA of ZO, claudin-1, occludin, connexin-43 ↑α-diversity ↑*Firmicutes*/*Bacteroidetes* ratio ↓*Alistipes*, *Helicobacter* ↓serum LPS, IL-18, TNF-α, IL-1β	[91]
ALP-1	Root of Burdock *Arctium lappa*	DSS-induced colitis in ICR mice Pretreat with ALP-1 (300 mg/kg) for 7 d followed by 3% DSS for 7 d	↑total gut bacteria ↑F/B ratio ↓*Proteobacteria* ↓*Staphyloccus* ↓colon and serum IL-1β, IL-6, TNF-α, ↑IL-10	[92]
ASPP	Purple sweet potato	DSS-induced colitis in ICR mice 400 mg/kg with 2.5% DSS for 7 d	higher total bacteria higher diversity ↑ *Firmicutes*/*Bacteroidetes* ratio, ↓ *Proteobacteria* ↑acetate, propionate ↓IL-1β, IL-6, TNF-α Changes in 25 functional pathways	[93]
WPSPP-1	Purple sweet potato	DSS-induced colitis in ICR mice 400 mg/kg with 4% DSS for 7 d	higher total bacteria ↑ *Firmicutes*/*Bacteroidetes* ratio, ↑acetate, propionate, butyrate ↓ *Proteobacteria* ↓IL-1β, IL-6, TNF-α, ↑IL-10	[94]
BSDF-1	Bamboo shoot *Phyllostachys edulis*	DSS-induced colitis in C57BL/6 mice Pretreat (100, 200, 400 mg/kg) for 7 d followed by 4% DSS for 7 d	↑ mRNA of occludens-1, claudin-1, occluding ↓*Parabaceroides*, *Mucispirillum* ↓*Helicobacter*, *Bacteroides* ↓*Streptococcus* ↑*Prevotella*, *Alitipes*, *Anaerostipes* ↑*Odoribacter*, *Bifidobacterium* ↑*butyricimonas*, *Lactobaccillus* ↓NF-κB, NLR	[95]
NFP	Noni fruit *Morinda citrifolia* L.	DSS-induced colitis in C57BL.6 mice 10 mg/kg with 2% DSS for 11 d	redistribution of ZO-1, occludin in colonic epithelial cells thicker mucus layer more goblet cells	[96]
HECP	Mushroom *Hericium erinaceus*	DSS-induced colitis in C57BL/6 mice 250, 500 mg/kg with 2% DSS for 7 d	↓*Verrucomicrobia* ↓*Actinobacteria* ↓amino acids metabolism pathways ↓NF-κB, Akt, MAPK ↓oxidative stress	[97]
FVP	Mushroom *Flammuliana velutipes*	DSS-induced colitis in Spragur Dawley rats Pretreat with 50, 100, 200 mg/kg followed by 4.5% DSS for 7 d	↑ α-diversity ↑ F/B ratio ↑*Ruminal butyrivibrios* ↑*Roseburia*, S24-7 ↓*Helicobacteraceae* ↑butyrate, isovaleric acid, valeric acids	[98]
DIP	Mushroom *Dictyophora indusiata*	DSS-induce colitis in BALB/c mice Pretreat with 10, 33 mg/kg DIP for 7 d followed by 3.5% DSS for 7 d	↑total gut bacteria ↑ F/B ratio ↓*Enterobacteriales* ↓*Gammaproteobacteria* ↓inflammatory cytokines ↓oxidative stress ↑mucins, goblet cells	[99]

## Data Availability

The study did not report any data.

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
