# Peer review of "Influence of Foods and Nutrition on the Gut Microbiome and Implications for Intestinal Health"

_ijms, 2022, doi:10.3390/ijms23179588_

Round 1
Reviewer 1 Report
Referee’s comment
Article n°: IJMS 1851731
Title: Influence of foods and nutrition on the gut microbiome and 2 implications for intestinal health
Authors: Ping Zhang
This review is almost interesting and well-organised. The argument treated should be of interest to scientist working in the field of nutrition as well as other s with closely related research interest. Therefore, this article collects the novelty of interest for the research community and in conclusion, considering the above matters, I retain that this manuscript could be suitable for publication in IJMS. However I have some suggestions to give more value to the manuscript and to attract further interest of the reader.
1) I think it is important to include a paragraph that considers the negative impact on the microbiome due to the indiscriminate use of antibiotics.
2) It would also be appropriate to add a paragraph relating new and current pharmacological approaches in this context.
3) It would also be appropriate to add a paragraph relating to innovative and future perspectives on the use on the microbiome transplant.
Reviewer 2 Report
The authors wrote a quite interesting review on food, nutrients and intestinal health and diseases. This is generally of high interest. This covers good amounts of data although it lacks discussion in some areas.
Diets certainly influence the microbiome, immune system, intestinal health, and pathogenic mechanisms. Diet can also interact with other factors, eg, drinks, water, beverages, alcohol, smoke, obesity, sleep, exercise, bowel habits, etc. These factors together may influence molecular pathologies in each patient differentially.
There are also influences of germline genetic variations on microbiome and cancer. Gene-by-environment interactions should be discussed.
One point that is missed in the current manuscript is that diets can be analyzed in relation to incidence (or course) of disease (such as IBD) but also pathogenic mechanisms in diseased tissue. In IBD, pathogenic mechanisms such as microbes and immune cells in tissue microenvironment can be examined. However, this type of approach is rarely taken. This is a widely open opportunity that is currently missed. Current common approaches are to examine foods / nutrients in relation to stool microbes; to examine foods / nutrients in relation to disease (eg, IBD); to examine stool microbes in relation to disease. Those are all piecemeals and not integrated.
In all of the above contexts, the authors discuss open opportunities such as research on dietary / lifestyle factors, microbiome, and personalized molecular biomarkers, which is needed for further research. The authors should discuss molecular pathological epidemiology research that can investigate diet, microbiota, and other factors in relation to molecular pathologies and clinical outcomes. Molecular pathological epidemiology research can be a promising direction (see Ann Rev Pathol 2019; Gut 2022, online ahead of print) and should be discussed in this paper.
